# AUTOMATIC MODERATOR DISCOVERY VIA SHAP INTERACTION VALUES

## ABSTRACT

Machine Learning (ML) is increasingly applied across the sciences, accelerating simulations, automating data preparation, and improving predictive accuracy. Yet most efforts emphasize efficiency and performance, with limited attention to interpretability, thereby leaving unexplored how ML can drive discovery—uncovering novel patterns in data and advancing scientific theory. Moderation effects—where the influence of one variable depends on the level of another—are central to disciplines such as social science and human behavior. However, they are typically studied through a theory-driven process based on regression models with manually specified interactions. While insightful, this approach is limited because it scales poorly and may miss unexpected moderators. We introduce an automated, interpretable framework for moderator discovery based on SHAP interaction values. Our method computes global interaction contributions from a predictive model, quantifies their dependence on constituent features, and identifies statistically significant moderators. In experiments on real-world datasets, the framework not only recovers known, theory-consistent moderating effects but also uncovers novel moderator candidates. These results illustrate how explainable ML can move beyond prediction toward systematic discovery, offering scientists a scalable tool to reveal conditional relationships that inform theory development.

## 1 INTRODUCTION

Machine learning (ML) has profoundly reshaped the practice of science in recent years. In what is often referred to as AI for science, ML methods have supported hypothesis generation, predictive modeling, and accelerated simulations of complex scientific phenomena, etc. Raghu & Schmidt (2020); Suresh et al. (2024); Eger et al. (2025); Šturek & Lazarova-Molnar (2025). For instance, ML-driven approaches have demonstrated impressive predictive power in genomics Jumper et al. (2021), materials science Butler et al. (2018), and neuroscience Bessadok et al. (2022), revealing structures and relationships that were previously inaccessible. Despite these advances, most existing ML-for-science efforts still emphasize predictive performance and computational gains. But interpretable insights and theory building remain underexplored.

Although there has been growing interest in interpretable machine learning to aid discovery—for example, enabling scientists to inspect black-box models and extract domain-meaningful patterns Wetzel et al. (2025); Allen et al. (2023); Anastasopoulos & Whitford (2019)—such efforts remain relatively niche and exploratory. This leaves a critical gap: how can ML help scientists not just label or cluster data, but *discover* and *interpret* emergent, novel patterns that then inform new theories?

One promising avenue toward theory-oriented discovery is to focus on the conditional nature of scientific relationships among variables. Across disciplines, researchers often turn to **moderating effects** to capture how the association between two variables may change depending on the context (moderator) Wu & Zumbo (2008).

**Moderating Effect** Consider a response variable $Y$, a focal predictor $X$, a candidate moderator $Z$, and a set of control variables $C$. A standard regression formulation with moderation can be written as:

$$Y = \beta_0 + \beta_1 X + \beta_2 Z + \beta_3 (X \times Z) + g(C) + \epsilon, \tag{1}$$

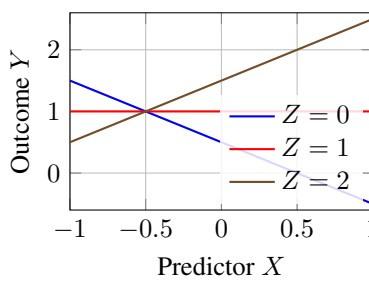 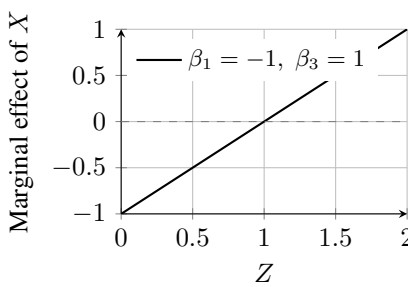

(a) Moderating effect of $Z$ on $X \rightarrow Y$.   (b) Marginal effect of $X$ as a function of $Z$.

Figure 1: Illustration of moderating effect: (a) Outcome $Y$ changes with $X$ across values of moderator $Z$, and (b) the marginal effect of $X$ increases linearly with $Z$.

where $\epsilon$ denotes a noise term, and $g(C)$ represents the contribution from control variables (which are included to adjust for confounding but are not the focus of moderation analysis). The coefficient $\beta_3$ captures the extent to which the effect of $X$ on $Y$ depends on $Z$ (Figure 1a).

The marginal effect of $X$ on $Y$ is therefore conditional on $Z$:

$$\frac{\partial Y}{\partial X} = \beta_1 + \beta_3 Z. \tag{2}$$

When $\beta_3 \neq 0$, the magnitude—and potentially the direction—of the $X \rightarrow Y$ association varies systematically with $Z$, indicating the presence of a *moderating effect*. Figure 1b illustrates a concrete example.

Moderators play a pivotal role in theoretical development across disciplines because they delineate the boundary conditions under which relationships between variables hold, weaken, or even reverse Hayes (2017); Hecht et al. (2023); Karataş & Cutright (2024); Krefeld-Schwalb et al. (2024). For example, in education research, the effect of study hours ($X$) on exam performance ($Y$) may depend on sleep quality ($Z$): students with higher sleep quality benefit more from additional study time. In this case, sleep quality serves as a **moderator**, altering the strength of the relationship between studying and performance.

Traditionally, however, the identification of moderators has been predominantly theory-driven, relying on domain expertise to specify contextual contingencies or dual-role variables. Despite their importance, explicit data-driven efforts to uncover moderators remain sparse and fragmented. Prior attempts have largely been domain-specific: for instance, Parr et al. (2022) applied random forests in meta-regression to screen for moderators in alcohol-use intervention studies, while Richter et al. (2025) used ML in a meta-analysis of cognitive training studies to highlight traits linked to heterogeneous intervention outcomes. These approaches demonstrate that ML can surface moderation-like patterns, but they remain ad hoc, prior-knowledge dependent, and problem-specific. What is still missing is a general, interpretable, and systematic framework capable of discovering and quantifying moderating effects across all variable pairs in high-dimensional settings.

In this work, we address this gap by proposing a comprehensive, interpretable, and scalable framework for data-driven moderator discovery. Grounded in SHAP interaction values, our approach moves beyond hypothesis-limited inquiry toward algorithmic discovery, thereby complementing theory-driven analyses and expanding the capacity for context-aware modeling. Concretely, our method proceeds in three steps: (1) train a high-capacity predictive model and compute SHAP interaction values for all feature pairs on a set of test instances; (2) evaluate the strength of potential moderation by quantifying how these interaction values vary with their constituent features; and (3) identify statistically significant moderators based on both effect magnitude and formal significance testing. This systematic, hypothesis-agnostic procedure enables researchers to uncover hidden moderation effects directly from data while preserving interpretability—advancing the role of explainable ML from predictive accuracy toward theory-oriented scientific discovery.

## 2 RELATED WORK

In this section, we first review domain-specific studies that use ML to uncover moderators in scientific contexts. We then turn to broader approaches aimed at detecting feature interactions in ML models.

### 2.1 DOMAIN-SPECIFIC APPROACHES FOR MODERATOR DISCOVERY

Recent studies demonstrate the utility of ML methods tailored to domain-specific moderator detection. For instance, Parr et al. (2022) proposed a two-stage framework in a meta-analysis of brief alcohol interventions, utilizing random forests to screen and rank candidate moderators before testing them in meta-regression. This approach highlighted factors such as prescriptive advice, comparison group type, and attrition as significant moderators. Similarly, Richter et al. (2025) leveraged random forests with SHAP analysis in an individual-level meta-analysis of cognitive training interventions for anxiety and depression, identifying moderators such as symptom severity and training dosage that influenced intervention efficacy. Extending beyond domain-specific applications, Feuerriegel et al. (2024) outlined how causal machine learning methods—such as causal forests—can estimate heterogeneous treatment effects from clinical trial and real-world data, thereby uncovering patient subgroups that moderate treatment efficacy. Along the same line, Zilcha-Mano et al. (2018) applied recursive partitioning with random forests to late-life depression trials, showing that education and illness duration interacted to differentially moderate the placebo versus medication response, revealing clinically meaningful subgroups that were otherwise obscured in aggregate analyses.

While these studies demonstrate the adaptability of ML for moderator discovery in diverse scientific contexts, they are typically highly customized, tailored to narrow application domains, and reliant on explicitly prespecified candidate moderators. Such constraints limit their scalability and generalizability, underscoring the need for systematic and interpretable frameworks for moderator discovery that can be applied across disciplines.

### 2.2 GENERAL APPROACHES FOR INTERACTION DETECTION

Beyond domain-specific applications, a growing body of research has explored feature interactions to understand ML model behavior. Early approaches relied on classical statistical tools, such as ANOVA or additive models with LASSO regularization, where interaction terms are explicitly constructed and irrelevant ones are shrunk toward zero Bien et al. (2013). While these approaches are theoretically grounded and straightforward to implement, they offer only limited interpretability without explaining the structured conditional relationships that characterize true moderating effects (Eq. 7).

Tree-based methods have long been proposed to address interaction discovery. For example, additive groves of trees detect interactions by examining variable splits across ensembles Sorokina et al. (2008), and model-based importance measures quantify the contribution of features and their pairwise interactions to prediction accuracy Greenwell et al. (2018). These methods exploit the hierarchical structure of trees, offering computational tractability and some interpretability. Causal tree methods extend this line of work to heterogeneous treatment effect estimation, partitioning the covariate space into subgroups with distinct responses and thereby uncovering potential moderators without requiring prespecified interaction terms Athey & Imbens (2016). However, even these approaches stop short of systematically characterizing the nature of moderating relationships, instead flagging subgroups or splits as potentially important.

Attribution-based methods provide yet another perspective. Extensions of Shapley values, such as the Shapley-Taylor interaction index Grabisch & Roubens (1999) and Integrated Hessians Janizek et al. (2021), quantify pairwise contributions via game-theoretic or gradient-based formulations. Similarly, Archipelago attributes interaction effects in neural networks using mixed partial derivatives Tsang et al. (2020). Recent work also emphasizes robustness: Li et al. (2023) introduced the feature interaction score cloud (FISC) to quantify variability of detected interactions across a Rashomon set of near-optimal models. While these methods are mathematically principled and model-agnostic, their discovery typically stops at a general interaction level, without providing a fine-grained analysis of how one variable conditions the effect of another on the outcome.

## 3 PROPOSED METHOD

We introduce a three-step framework for systematic moderator discovery from observational data. Let $Y$ denote the response variable and $X = \{X_1, \ldots, X_M\}$ the set of predictors. Our goal is to identify variables that moderate the effect of one predictor on the outcome by leveraging SHAP interaction values. The procedure consists of three main steps: 1) **Model Training**, 2) **Interaction Attribution**, 3) **Moderator Assessment**.

The framework is designed to be hypothesis-agnostic. Unlike classical regression-based moderation analysis, which requires researchers to prespecify candidate moderators, our approach leverages machine learning to screen for moderation systematically across all possible feature pairs. By building on SHAP interaction values, we retain interpretability while handling high-dimensional data in a principled manner.

### 3.1 MODEL TRAINING: GRADIENT-BOOSTED DECISION TREES

The first step is to train a high-capacity model that is capable of capturing complex relationships between predictors and response variables. Gradient-boosted decision-tree (GBDT) ensembles Friedman (2001) combine strong predictive accuracy with model-specific interpretability. They construct an additive model by sequentially fitting CART-style regression trees, which capture nonlinearities and higher-order interactions. GBDTs are particularly well-suited to scientific data, where predictors may be heterogeneous (continuous, categorical, ordinal) and relationships are nonlinear.

**Why GBDTs for moderation**  Several properties make GBDTs ideal for our setting. First, they naturally partition the feature space into regions where interactions may differ, aligning well with the concept of conditional or context-specific effects. Second, they avoid the need to explicitly enumerate interaction terms, which can be a bottleneck in classical regression. Third, GBDTs remain competitive with deep learning models in structured data while being more interpretable and easier to audit.

**TreeSHAP efficiency**  Crucially, the TREESHAP algorithm Lundberg et al. (2020) computes exact SHAP values (including interaction values) for tree ensembles in polynomial time with respect to tree depth and linear time with respect to tree size. This means that even for large ensembles trained on thousands of features, SHAP interaction values can be computed efficiently, enabling large-scale moderator discovery.

**LightGBM Backbone**  Let $\mathcal{D} = \{(\boldsymbol{x}^{(n)}, y^{(n)})\}_{n=1}^N$ denote a dataset with $N$ instances, where each feature vector is $\boldsymbol{x}^{(n)} = (x_1^{(n)}, \ldots, x_M^{(n)}) \in \mathbb{R}^M$ and the response variable is $y^{(n)} \in \{0, 1\}$. A gradient-boosted decision tree (GBDT) model learns an additive ensemble of $T$ trees,

$$\hat{y}^{(n)} = f(\boldsymbol{x}^{(n)}) = \sum_{t=1}^T g_t(\boldsymbol{x}^{(n)}), \tag{3}$$

where each $g_t \in \mathcal{G}$ is a CART-style regression tree mapping $\mathbb{R}^M \to \mathbb{R}$.

In principle, any GBDT implementation, such as XGBoost Chen & Guestrin (2016) or CatBoost Prokhorenkova et al. (2018), could serve as the predictive backbone for our framework. We adopt LIGHTGBM Ke et al. (2017) primarily for its simplicity, ease of use, and seamless compatibility with TREESHAP. In addition, LightGBM scales efficiently to large datasets via histogram-based feature binning and leaf-wise growth while maintaining strong predictive performance. These properties make it a practical and effective choice for our moderator discovery framework; however, the method itself is not limited to this implementation.

### 3.2 INTERACTION ATTRIBUTION: SHAP INTERACTION MATRIX

Having trained a predictive model, the next step is to extract interaction information in a way that is faithful to the fitted model and interpretable to researchers.

For a given instance $\boldsymbol{x}^{(n)} \in \mathbb{R}^M$, TREESHAP returns an $M \times M$ interaction matrix

$$\boldsymbol{\Phi}(\boldsymbol{x}^{(n)}) = \left[ \phi_{i,j}(\boldsymbol{x}^{(n)}) \right]_{i,j=1}^M, \qquad \phi_{i,j}(\boldsymbol{x}^{(n)}) = \phi_{j,i}(\boldsymbol{x}^{(n)}). \tag{4}$$

The standard (first-order) SHAP value for feature $X_j$ is recovered as

$$\phi_j(\boldsymbol{x}^{(n)}) = \sum_{i=1}^M \phi_{i,j}(\boldsymbol{x}^{(n)}). \tag{5}$$

To focus explicitly on interactions, we define the joint attribution of two features $X_i$ and $X_j$, for instance $n$ as

$$\phi_{i,j}^{\text{tot}}(\boldsymbol{x}^{(n)}) = \phi_{ii}(\boldsymbol{x}^{(n)}) + \phi_{jj}(\boldsymbol{x}^{(n)}) + 2\,\phi_{ij}(\boldsymbol{x}^{(n)}). \tag{6}$$

This quantity represents the total contribution of the pair $\{X_i, X_j\}$ to the prediction for instance $n$, combining their main effects and their interaction. By aggregating across all instances, we obtain a sample-level view of how feature pairs behave, setting the stage for moderation analysis.

### 3.3 MODERATOR ASSESSMENT: MARGINAL AND MODERATING EFFECTS

The final step is to determine whether one feature serves as a moderator of another. Intuitively, we ask: does the contribution of $X_i$ to the prediction depend systematically on the value of $X_j$?

**Regression-based assessment**  We identify moderators by linking joint attributions to feature values through a regression model:

$$\phi_{i,j}^{\text{tot}}(\boldsymbol{x}^{(n)}) \approx \beta_0 + \beta_1 x_i^{(n)} + \beta_2 x_j^{(n)} + \beta_3\, x_i^{(n)} x_j^{(n)}. \tag{7}$$

Here, $\beta_3$ is the *moderation coefficient*, quantifying whether the contribution of $X_i$ depends systematically on the value of $X_j$. The regression formulation mirrors classical moderation analysis in social science, but crucially, our dependent variable is not $Y$ itself but the SHAP-derived contribution of the feature pair to the model's prediction. This design bridges the interpretability of regression with the predictive capacity of ML models. Additionally, for fair comparison between different interactions, we normalize all predictors before the regression.

**Interpretation**  The coefficient $\beta_3$ captures the direction of moderation. A positive value indicates that $X_j$ amplifies the effect of $X_i$, a negative value means $X_j$ attenuates or reverses it, and a value near zero suggests no meaningful moderation.

**Statistical testing**  For each feature pair $(i, j)$, we estimate $\beta_3$ in Eq. equation 7 and assess its significance using standard $t$-tests. This ensures that detected moderators reflect meaningful conditional relationships rather than random variation. In practice, we also report effect sizes to distinguish statistically significant but practically negligible moderators.

### 3.4 SUMMARY OF THE FRAMEWORK

In summary, our framework combines three components (Algorithm 1): (1) high-capacity predictive modeling via LightGBM, (2) principled interaction attribution using SHAP values, and (3) regression-based moderation assessment with statistical testing.

This design provides a scalable and interpretable pipeline for systematic moderator discovery. By grounding the analysis in model-derived contributions rather than raw outcomes, the approach enables researchers to uncover hidden conditional relationships while preserving the rigor of statistical inference. Ultimately, the framework extends explainable ML beyond predictive accuracy, aligning it with the scientific goal of theory generation and boundary condition discovery.

---

**Algorithm 1** Automatic Moderator Discovery via SHAP Interaction Values

---

**Require:** Dataset $\mathcal{D}$ with $N$ samples and $M$ features; significance level $\alpha$
**Ensure:** Ranked list of significant moderator pairs $(i, j)$
 1: **Model Training:** Train LightGBM model $f$ as in Eq. 3.
 2: **Interaction Attribution:** For each instance, compute SHAP interaction matrix Eq. 4. For each pair $(i, j)$, compute joint attribution using Eq. 6.
 3: **Moderator Assessment:** Fit regression Eq. 7 for each $(i, j)$; extract $\hat{\beta}_3$ and $p$-value from a $t$-test. Retain all $(i, j)$ with $p < \alpha$, ranked by $|\hat{\beta}_3|$.

---

## 4 EXPERIMENT

**Baselines** Most existing works focus on general interaction detection rather than moderating effect discovery. Therefore, we compare our framework against the following representative baselines for interaction detection:

- **Hierarchical Lasso** Bien et al. (2013): An extension of the standard lasso that imposes convex hierarchy constraints, ensuring that an interaction term $X_i X_j$ is included only if at least one of the corresponding main effects $X_i$ or $X_j$ is selected. This structured sparsity approach identifies relevant interactions by shrinking irrelevant coefficients to zero while maintaining interpretability through hierarchical feature selection.

- **Shapley Interaction Index** Muschalik et al. (2024); Grabisch & Roubens (1999): A cooperative game–theoretic measure that quantifies the joint contribution of feature subsets by averaging discrete derivatives across all coalitions. This provides a principled measure of interaction strength but is agnostic to directionality or conditioning, and therefore does not explicitly capture moderation effects. In our experiments, the Shapley Interaction Values are obtained the same way as our approach, i.e., first train a LightGBM model then extract interaction values via TreeSHAP.

- **ANOVA** Montgomery (2017): The classical two-way analysis of variance, which tests for interaction effects by evaluating the statistical significance of product terms (e.g., $X_i \times X_j$) in a linear model. This provides a traditional hypothesis-testing baseline for detecting non-additive relationships between predictors.

**Variants of Our Method** In addition to comparing against external baselines, we consider two internal variants of our framework to isolate the contribution of SHAP interaction values. The first variant, **ShapMod-1st**, replaces Eq. 6 with first-order SHAP values only, i.e., marginalizing interaction terms:

$$\phi_{i,j}^{\text{tot}}(\boldsymbol{x}^{(n)}) = \phi_i(\boldsymbol{x}^{(n)}) + \phi_j(\boldsymbol{x}^{(n)}).$$

The second variant, **ShapMod**, is our full method, which leverages both main effects and pairwise SHAP interaction values as defined in Eq. 6.

**Datasets** We include one synthetic dataset and two real-world datasets in social science to validate our framework.

- **Synthetic Data**: We generate 3000 samples from a model of 40 variables with 4 interactions:

$$y = \sum_{i=1}^{40} x_i + \sum_{i=1}^{4} x_i x_{i+1}.$$

  The goal is to successfully retrieve the 4 interactions using the proposed approach.

- **Recidivism Prediction** (COMPAS): The COMPAS dataset Flores et al. (2016) contains criminal history, demographics, and charge information for defendants, and is widely used to study fairness and predictive accuracy in recidivism risk assessment. We apply our method to examine interaction effects underlying recidivism outcomes.

- **Vaccination Uptake**: This dataset investigates determinants of vaccination rates among Medicare beneficiaries across racial groups. It includes contextual factors such as education, income, social capital, and community health, allowing us to explore how structural conditions shape vaccination disparities Peng & Yang (2025).

**Evaluation Metrics**   We evaluate discovered moderators by testing whether the interaction coefficient is statistically significant when added to a linear model. For each candidate moderator pair $(X_i, X_j)$, we fit the following regression:

$$Y = \beta_0 + \sum_{k=1}^{M} \beta_k X_k + \beta_{ij} (X_i \times X_j) + \epsilon,$$

where $\beta_{ij}$ quantifies the moderating effect of $X_j$ on $X_i$. The estimated coefficient $\hat{\beta}_{ij}$ captures the direction and magnitude of moderation, while its $p$-value assesses statistical significance under the null hypothesis $\beta_{ij} = 0$. We treat an interaction as a *hit* if $p < \alpha$ (default $\alpha = 0.01$).

To compare different discovery methods, we generate cumulative significance curves based on the ranked list of candidate interactions. For the top-$K$ pairs proposed by a method, we compute two curves as a function of interaction rank $t$:

$$H(t) \;=\; \sum_{k=1}^{t} \mathbf{1}\{p_{i_k j_k} < \alpha\}, \qquad R(t) \;=\; \frac{H(t)}{t}.$$

Here $H(t)$ is the *cumulative hit count*, showing how many significant moderators have been identified up to rank $t$, and $R(t)$ is the *cumulative hit ratio*, showing the proportion of significant moderators among the top-$t$ candidates. Together, these curves provide a rank-sensitive view of each method's performance of locating moderating effects.

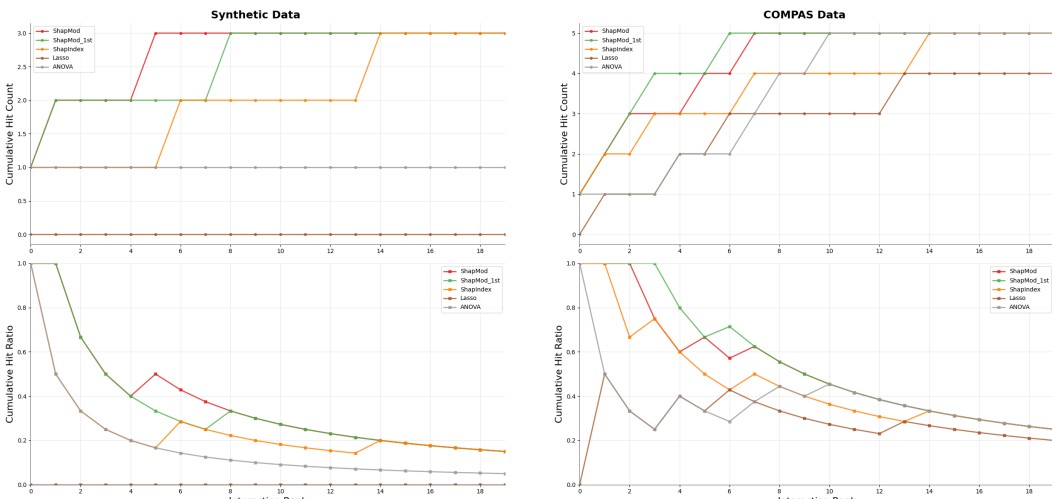

Figure 2: Results on synthetic data (left panel) and COMPAS data (right panel). A good moderator detection approach should recover significant interactions as early as possible, which is reflected in the plots as the best curve should be above other methods.

### 4.1 SYNTHETIC DATA

Figure 2 (left) presents results on the synthetic dataset, where only a small number of interactions present and inducing moderating effects. A good moderator detection approach should recover significant interactions as early as possible, which reflect in the plots as the best curve should locate above all other methods.

In this setting, **ShapMod** and **ShapMod-1st** clearly outperform all other baselines. The cumulative hit count/ratio curves show that these methods surface the majority of significant moderators within the top few ranked candidates.

## 4.2 RECIDIVISM ANALYSIS (COMPAS DATASET)

Figure 2 (right) presents results on the COMPAS dataset, it shows similar tendency to favor our approach.

More specifically, we retrieve the top-ranked moderators of recidivism risk, including age, race, prior criminal history, and juvenile records (Table 1). Our results align with established sociological and criminological theories.

Table 1: Top-ranked interactions in the COMPAS dataset retrieved by our ShapMod approach. Here, *juvenile_misdemeanor_count*, *priors_count*, *juvenile_other_count*, and *juvenile_felony_count* denote counts of different types of prior misbehaviors.

| Feature $i$ | Feature $j$ | $\beta_3$ |
|---|---|---|
| age | juvenile_misdemeanor_count | -0.1136 |
| age | priors_count | -0.0871 |
| age | race | -0.0731 |
| age | juvenile_other_count | -0.0678 |
| race | priors_count | -0.0557 |
| age | juvenile_felony_count | -0.0541 |
| age | charge_degree | -0.0470 |
| juvenile_misdemeanor_count | priors_count | -0.0406 |
| priors_count | charge_degree | -0.0364 |
| juvenile_felony_count | priors_count | -0.0221 |

In particular, labeling theory helps explain why prior criminal record consistently emerges as one of the most influential moderators. Once individuals are officially labeled as "criminals," they face stigmatization, diminished employment prospects, and restricted opportunities for reintegration into society—effects that are especially severe for Black ex-offenders. This dynamic produces a "double stigma," where racial bias amplifies the negative consequences of criminal labeling, perpetuating cycles of disadvantage and reoffending Bontrager et al. (2005); Chiricos et al. (2007).

In addition, the repeated presence of age in interaction terms reflects life-course theory (Wojciechowski, 2025; Uggen, 2000). Younger offenders with prior juvenile misdemeanors or felonies are often perceived as more risky and less likely to desist, consistent with criminological perspectives emphasizing the elevated recidivism risk of youthful repeat offenders. Age also interacts with criminal history: even offenders with extensive records show lower predicted risk at older ages, a pattern consistent with desistance theory.

Thus, our method not only validates theoretical expectations but also quantifies how age moderates the influence of prior and juvenile criminal history in shaping recidivism outcomes, bridging data-driven discovery with theory-driven interpretation.

## 4.3 VACCINATION DATA ANALYSIS

Table 2: Top-ranked interactions retrieved by our ShapMod approach for vaccination rates in different racial groups.

| (a) White beneficiaries | | | (b) Black beneficiaries | | |
|---|---|---|---|---|---|
| Feature $i$ | Feature $j$ | $\beta_3$ | Feature $i$ | Feature $j$ | $\beta_3$ |
| Population | Internet | -0.0125 | Social Capital | Poor Health | 0.0070 |
| Population | Education | -0.0078 | Social Capital | Internet | -0.0060 |
| Social Capital | Internet | -0.0067 | Residential Segregation | Poor Health | 0.0058 |
| Social Capital | Political Ideology | -0.0065 | Community Resilience | Elder Population | 0.0046 |
| Political Ideology | Population | -0.0064 | Social Capital | Residential Segregation | -0.0044 |
| Social Capital | Income | -0.0063 | Community Resilience | Poor Health | 0.0044 |
| Social Capital | Education | -0.0060 | Social Capital | Political Ideology | -0.0043 |
| Population | Income | -0.0058 | Social Capital | Education | -0.0041 |
| Political Ideology | Education | -0.0052 | Social Capital | Income | -0.0041 |
| Population | Residential Segregation | -0.0049 | Social Capital | Community Resilience | -0.0038 |

We applied our method to Medicare vaccination data (Figure 3), which investigates the determinants of vaccination uptake across racial groups. Specifically, we used our approach to two racial groups:

White beneficiaries (Figure 3 left) and Black beneficiaries (Figure 3 right). For White beneficiaries, the top-ranked moderators (Table 2a) include interactions between community social capital and key contextual variables such as education, median household income, internet access, political ideology, and population size. For Black beneficiaries (Table 2b), we observed a partially overlapping but distinct interaction pattern, in which social capital remained central but was now combined with community poor health, residential segregation, and the proportion of older adults. These findings highlight social capital as a recurrent moderator across groups, though the secondary interacting factors differ, reflecting distinct structural conditions.

From a theory-driven perspective, public health studies have discussed the contextual role of social capital in health and health equity Wilkinson (2002); Uphoff et al. (2013). Empirical studies further demonstrate its moderating effects. For instance, Zhang et al. (2022) show that political ideology moderates the effect of social capital on COVID-19 vaccination rates, while Jung et al. (2013) demonstrate that social capital interacts with parental knowledge in shaping childhood vaccine uptake. Our findings, which consistently show that social capital emerges as a significant moderator, provide data-driven support for these theoretical perspectives.

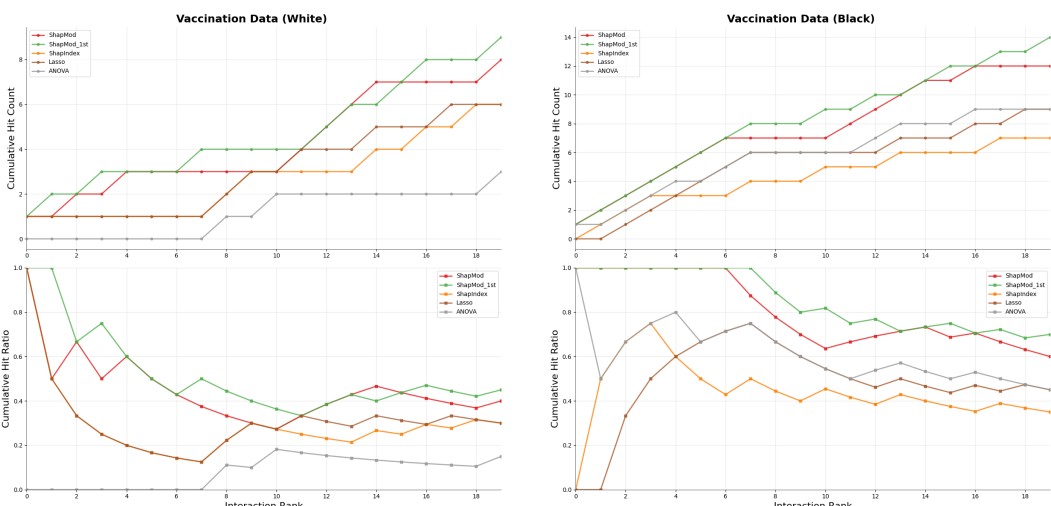

Figure 3: Vaccination dataset for white population (left) and black population (right).

Beyond established moderators, our method also uncovered novel patterns, particularly for Black beneficiaries. Residential segregation emerged as a strong interacting factor with both social capital and community health burdens. While prior studies have primarily emphasized the direct effects of segregation on healthcare access and resource distribution (Anderson & Ray-Warren, 2022; Strully, 2011), our findings extend this perspective by demonstrating that segregation interacts with other contextual factors. These segregation-related interactions reveal mechanisms that go beyond traditional factors and directly implicate structural inequities in shaping vaccination disparities.

## 5 CONCLUSION

We presented a framework for automatic moderator discovery using SHAP interaction values, moving beyond general interaction detection toward identifying conditional relationships that inform theory development. Across synthetic and real-world datasets, our method consistently recovered known moderators and outperformed classical baselines in sparse and structured settings. While our evaluation currently relies on linear regression to test significance—thus simplifying the treatment of control variables—future work should extend this to more flexible models that capture nonlinear or complex dependencies. Overall, our results highlight the promise of explainable ML as a scalable and interpretable tool for systematic moderator discovery in scientific research.

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
