# OpenReview forum: "Automatic Moderator Discovery via SHAP Interaction Values"
_ICLR.cc/2026/Conference — Submitted to ICLR 2026_

### Official Review · Reviewer_wtgj · 2025-10-17

**Soundness:** 2
**Presentation:** 2
**Contribution:** 2
**Rating:** 2
**Confidence:** 3

**Summary:**

The paper proposes an automated framework for data-driven moderator discovery using SHAP interaction values derived from gradient-boosted decision trees (LightGBM). The framework aims to bridge predictive ML and theory-driven science, allowing researchers to systematically uncover moderating effects (conditional relationships) without prespecifying candidate moderators.

**Strengths:**

### **Clear Motivation**
* The work addresses an underexplored gap between predictive modeling and interpretable, theory-oriented discovery, particularly relevant in social sciences and behavioral domains.

### **Methodology**
* Leveraging TreeSHAP for interaction attribution is clever: it enables scalable, model-faithful extraction of interaction information with polynomial-time complexity.

### **Empirical Validation**
* Experiments include both synthetic (ground-truth known) and real-world (social science) datasets.

**Weaknesses:**

### **Technical Limitations / Mathematical Depth**
* The paper does not provide theoretical justification or bounds on when SHAP interaction values can be reliably interpreted as moderation effects. SHAP interaction measures contribution, not necessarily conditional effect in a causal or statistical moderation sense.
* The $\beta_3$ estimation step treats SHAP values as regression targets without addressing their statistical dependence structure or the implications of SHAP value non-orthogonality. This may lead to inflated significance or misinterpretation.
* No identifiability or robustness analysis is provided: for instance, how stable the discovered moderators are to model perturbations, feature scaling, or correlated covariates.

### **Novelty**
* The method is essentially a combination of existing components: TreeSHAP, classical moderation regression, and significance testing
* Attribution-based moderation inference has conceptual similarity to existing feature interaction explanation methods

### **Conceptual Ambiguity**
* The paper equates SHAP interaction values with moderating effects, but this is not strictly justified: SHAP interactions are symmetrical, whereas moderation is directional (one variable moderates another). Although regression on SHAP attributions introduces directionality, the interpretation remains heuristic, not theoretically guaranteed.

**Questions:**

Please see the above weaknesses

---

> ### Author Response · Authors · 2025-12-04
>
> 1. The paper does not provide theoretical justification or bounds on when SHAP interaction values can be reliably interpreted as moderation effects. SHAP interaction measures contribution, not necessarily conditional effect in a causal or statistical moderation sense.
>
> Our framework is designed as a **Discovery Tool** rather than a causal oracle. We provide statistical guarantees regarding the significance of the interaction candidates (via statistical significance), but the final theoretical validation of the "moderator" role must involve the domain user.
>
> 2. The $\beta_3$ estimation step treats SHAP values as regression targets without addressing their statistical dependence structure or the implications of SHAP value non-orthogonality. This may lead to inflated significance or misinterpretation.
>
> The SHAP values serve as disentangled attributions, we estimate  $\beta_3$ by only incorporating relevant SHAP values ($\phi_i,\phi_j,\phi_{ij}) , which implicitly rules out the statistical dependence structure with other features.
>
> 3. No identifiability or robustness analysis is provided: for instance, how stable the discovered moderators are to model perturbations, feature scaling, or correlated covariates
>
> We pre-normalize all variables to avoid the results being altered solely by feature scaling. Regarding robustness to model perturbations and covariate correlation, we acknowledge that the current version does not yet include a full sensitivity or stability analysis. We agree that these aspects are valuable, and we will incorporate them in the next revision.
>
> 4. The method is essentially a combination of existing components: TreeSHAP, classical moderation regression, and significance testing
>
> Our goal is not to introduce a new SHAP algorithm or a new statistical estimator, but rather to propose a new conceptual framework that connects model-based interaction attribution with theory-driven moderation analysis. While the individual components are indeed established tools, their integration for systematic, directional moderator discovery is novel and, to our knowledge, has not been explored in prior work on feature interaction or explainability.
>
> 5. Attribution-based moderation inference has conceptual similarity to existing feature interaction explanation methods
>
> As discussed in our Related Work (Section 2.2), the most relevant are general interaction detection methods (such as Integrated Hessians (Janizek et al., 2021), Archipelago (Tsang et al., 2020), and FISC (Li et al., 2023)) focus on identifying that an interaction exists . However, these methods typically stop at the "general interaction level" and do not provide a fine-grained analysis of the conditional relationship.
>
> 6. The paper equates SHAP interaction values with moderating effects, but this is not strictly justified: SHAP interactions are symmetrical, whereas moderation is directional (one variable moderates another). Although regression on SHAP attributions introduces directionality, the interpretation remains heuristic, not theoretically guaranteed.
>
> We would like to clarify that our method does not equate SHAP interactions with moderating effects. Instead, SHAP interaction values serve solely as model-faithful indicators of how the joint contribution of two variables varies across the data, providing a structured signal for subsequent analysis.
>
> We acknowledge that this approach is heuristic rather than a causal or theoretical guarantee, and we do not claim SHAP-derived moderation coefficients have the same formal identifiability properties as traditional parametric models. Instead, the framework is intended as a screening and discovery tool, enabling researchers to identify candidate conditional relationships that warrant further domain-specific investigation.

---

### Official Review · Reviewer_yJu9 · 2025-10-30

**Soundness:** 2
**Presentation:** 3
**Contribution:** 2
**Rating:** 4
**Confidence:** 3

**Summary:**

This paper proposes a three-step interpretable framework for the automatic discovery of moderator effects from data. The method aims to bridge predictive machine learning with theory-driven scientific discovery. The framework first trains a high-capacity GBDT model to capture complex relationships. Second, it computes pairwise SHAP interaction values to quantify the joint contribution of feature pairs. Third, it regresses these attribution scores onto the original feature values (fitting $\phi_{i,j}^{tot} \approx \dots + \beta_3 x_i x_j$) and uses the resulting $\beta_3$ coefficient and its p-value to identify and rank significant moderators. The method is validated on synthetic data and two real-world datasets demonstrating it can recover theoretically-consistent moderators and outperform baselines.

**Strengths:**

1. The paper addresses a highly important and practical goal: moving ML from a purely predictive tool to one that can aid in systemic, data-driven scientific discovery.
2. The paper is exceptionally well-written. The problem is clearly defined, the method is presented in a logical, three-step algorithm, and the experimental setup is easy to follow.
3. While the individual components (GBDT, SHAP, OLS) are standard, their composition is novel. The core original idea is to re-purpose SHAP values as an intermediate data source for a second-stage statistical analysis. This is a clever approach that reframes the vague problem of "interaction detection" into the more specific, interpretable problem of "moderator assessment."

**Weaknesses:**

1. The designed framework can only detect 2-way interactions. It is blind to higher-order interactions, which are common and critical in many scientific domains. This makes the contribution feel more like a proof-of-concept than a complete discovery system.
2. The contribution is a clever pipeline of existing tools rather than a new algorithm or fundamental theory. This limits the technical originality of the work.
3. The claim of a general, systematic framework is supported by only one synthetic and two real-world datasets. This is not extensive enough to prove its general applicability across different scientific domains.
4. The validation metric (a linear test) seems mismatched with the non-linear GBDT model.
5. The paper does not analyze how sensitive the discovered moderators are to the GBDT's hyperparameters, which is critical for a discovery tool's reliability.

**Questions:**

1. What are the primary conceptual or computational barriers to extending this framework to 3-way (or higher-order) interactions? Would this involve computing 3rd-order SHAP values and regressing them onto a term like $\beta x_i x_j x_k$?
2. Why use a linear test to validate a non-linear model? Doesn't this risk penalizing true non-linear discoveries? Relatedly, how sensitive is the final list of moderators to the GBDT's hyperparameters? We need to be sure the discoveries are robust and not artifacts of tuning.
3. The claim of a general framework is supported by only two real-world datasets. Could the authors comment on the expected generalizability to other scientific domains and data types? What challenges in applying this method to much higher-dimensional data (e.g., M > 10,000)?

---

> ### Author Response · Authors · 2025-12-04
>
> We thank the reviewer for the insightful comments and constructive suggestions. The concerns largely fall into three categories: (i) scope of the proposed framework, (ii) robustness and model-validation choices, and (iii) generalizability. We clarify that our goal is not to introduce a new learning algorithm, but to propose a general and interpretable framework that repurposes SHAP interaction values for statistical moderation analysis — a capability not offered by existing interaction-detection methods. We address each point below.
>
> 1. The designed framework can only detect 2-way interactions.
>
> We appreciate the reviewer pointing out the current focus on 2-way interactions. Our goal in this work is to provide a practical framework for moderator discovery, aligned with how moderation is defined and used in scientific domains (e.g., social sciences, public health, psychology). In these fields, two-way moderation effects are by far the dominant and most interpretable form, and they constitute the standard building block for theory development Thus, restricting our initial design to 2-way moderators matches both interpretability requirements and prevailing scientific practice.
>
> 2. The contribution is a clever pipeline of existing tools rather than a new algorithm or fundamental theory. This limits the technical originality of the work.
>
> While our framework indeed leverages established components, the technical originality lies in that it reframes interaction detection into moderator discovery — a task fundamentally distinct from existing attribution or interaction methods.
>
> 3. The claim of a general, systematic framework is supported by only one synthetic and two real-world datasets. This is not extensive enough to prove its general applicability across different scientific domains.
>
> We agree with the reviewer that the current empirical coverage is not as extensive as ideal. Our goal in this submission was to demonstrate the framework’s feasibility rather than exhaust every possible domain. Nonetheless, we acknowledge that additional experiments would strengthen the generality claims.
>
> 4. The validation metric (a linear test) seems mismatched with the non-linear GBDT model.
>
> The linear regression is not a test, it is a regression-based assessment to rank the moderator candidates.
>
> 5. The paper does not analyze how sensitive the discovered moderators are to the GBDT's hyperparameters, which is critical for a discovery tool's reliability.
>
> We agree with the reviewer that assessing sensitivity to GBDT hyperparameters is important for ensuring the robustness of a discovery tool. In this submission, we did not iterate over the full hyperparameter space, and we acknowledge that a more systematic sensitivity analysis would strengthen the reliability claims of the framework.

---

### Official Review · Reviewer_bQgQ · 2025-11-01

**Soundness:** 2
**Presentation:** 2
**Contribution:** 2
**Rating:** 2
**Confidence:** 2

**Summary:**

This paper introduces a scalable and interpretable framework for automatic moderator discovery in machine learning models. It uses SHAP interaction values to screen for and quantify moderating effects, where the influence of one variable depends on the level of another, across all feature pairs in high-dimensional data. Evaluation on synthetic and real-world datasets (COMPAS, Medicare vaccination) demonstrates the method’s ability to recover known moderators and reveal new candidates not previously highlighted by domain-driven analyses.

**Strengths:**

- The paper tackles an interesting but underexplored facet of interpretable machine learning.
- The methodology is well-motivated and clearly described. The use of SHAP interaction values for model-wide screening, combined with a regression-based moderation assessment, is a compelling merge of machine learning and statistical theory

**Weaknesses:**

1. While the paper evaluates the statistical significance of discovered moderators via inclusion in a linear regression, this does not fully control for potential confounding or capture non-linear relationships. The approach essentially reduces the problem to significance of pairwise terms, which may inflate discovery of spurious moderators, especially in complex or collinear datasets. Further, no detailed ablation/sensitivity studies are reported regarding how methodological or hyperparameter changes (e.g., choice of model class, depth of GBDT, or normalization scheme) affect results or stability of discovered moderators.
2.  The method is only tested using GBDT (LightGBM). While it is stated that other tree ensembles (XGBoost, CatBoost) or potentially other model classes could be used, no empirical evidence or discussion is provided for non-tree models, or for cases where SHAP values are approximated rather than computed exactly.
3. While the technique is positioned as “interpretable” and “scalable,” the practical interpretability of discovered moderators is not rigorously evaluated beyond recovering known scientific patterns. For instance, how do domain experts interpret SHAP-ranked moderators in highly collinear data, or when moderators do not have a causal interpretation?
4. The quality of Figure 2 could be improved. The legend/axes titles are too small, and there are no error bars/bands.

**Questions:**

1. Could the authors clarify the exact normalization approach for predictors used prior to the regression-based moderation test (Section 3.3)? For continuous vs. categorical features, is normalization performed identically?
2. How does the method perform if the predictive model is replaced with a neural network backbone, or if SHAP values must be estimated empirically rather than via TreeSHAP? Is the approach robust to approximate SHAP implementations?
3. How scalable is the method in practice as the number of features increases into the hundreds/thousands? Actual running time benchmarks or empirical scaling curves would be valuable for practitioners.

---

> ### Author Response · Authors · 2025-12-04
> **Thanks for the valuable feedbacks!**
>
> 1. Regression-based significance testing may not control for confounding or nonlinear relationships. How do domain experts interpret SHAP-ranked moderators in highly collinear data, or when moderators do not have a causal interpretation?
>
> We appreciate this insightful point. Our framework is designed as a **Discovery Tool** rather than a causal oracle. The purpose is to select a list of candidates for moderators. Although there might be confounders, it is always necessary to incorporate a final theoretical validation by the domain user.
>
> 2. Method is only tested with GBDT. It’s unclear whether it works with neural networks or approximate SHAP.
>
> We select GBDT mainly because we can relatively efficiently obtain SHAP values by TreeSHAP. In our scope of studies, such as datasets of 1k - 10k samples, a tree model is already powerful enough.
>
> 3. Could the authors clarify the exact normalization approach for predictors used prior to the regression-based moderation test (Section 3.3)? For continuous vs. categorical features, is normalization performed identically?
>
> We use standard normalization for variables $(X - \bar{X})/\sigma$, for categorical features, we convert them to one-hot features.
>
> 4.  How does the method perform if the predictive model is replaced with a neural network backbone, or if SHAP values must be estimated empirically rather than via TreeSHAP? Is the approach robust to approximate SHAP implementations?
>
> The GBDT is sufficiently enough in our study scenarios, and treeSHAP calculate exact SHAP values efficiently. If we use neural network, there might be overfitting issue as sample sizes are small. Furthermore, for NNs, the calculation of exact SHAP is extremely expensive, which may not be a viable approach. Instead, if we approximate SHAP, the quality may not be as reliable as treeSHAP.
>
> 5. How scalable is the method in practice as the number of features increases into the hundreds/thousands? Actual running time benchmarks or empirical scaling curves would be valuable for practitioners.
>
> We thank the reviewer for pointing this out. The scalability we mention in the paper is that by using our framework, scientist can verify candidate moderators more efficiently, instead of verifying all possible interactions, which is explosively large. That said, in our study scenarios, our method will always bottlenecked by the TreeSHAP algorithm.

---

### Official Review · Reviewer_3ihw · 2025-11-01

**Soundness:** 2
**Presentation:** 3
**Contribution:** 2
**Rating:** 2
**Confidence:** 4

**Summary:**

The paper presents a heurstic framework for automated, data-driven discovery of moderation effects (conditional dependence bet two variables given a context) using SHAP interaction values derived from gradient-boosted decision trees. The idea is well motivated and the empirical experiment is supported with domain insights, yet the methodology sound limited, please see comments below. Overall, I believe the scope and implementation of this work are of limited interest to ICLR community.

**Strengths:**

- Proposes a systematic approach for uncovering moderating effects. The approach is hypothesis-agnostic, scalable to high-dimensional datasets, and preserves model interpretability. Empirical results on synthetic and real data show that some known moderators and even novel candidates can be recovered.

**Weaknesses:**

- The use of GBDTs for moderator discovery indeed captures nonlinear interactions; but also results in discontinuous, piecewise-constant outputs. This discretization may not be able to capture smooth moderation patterns, limiting method generalization.

- The method computes SHAP interaction values via TreeSHAP (as indicators of moderation). TreeSHAP relies on sampling features from their product-marginal (i.e., assuming independence bet features) when estimating the contributions. Thus, it may ignore or misrepresent the true joint dependence structure among features, which can lead to artificial conditional relationships or spurious moderators being detected.

- ShapMod-1st outperforms ShapMod (based in Eq. 6) in three of four datasets, raising a question about the need for SHAP interaction terms in Eq (6)?

- Both the SHAP method and trained model introduce potential errors through estimation and initialization. Evaluation relies on small datasets, including a synthetic example with only 3,000 samples but many features (40), raising various questions about generalizability and sample-size sensitivity.

**Questions:**

See weaknesses section

---

> ### Author Response · Authors · 2025-12-04
> **Thanks for the valuable feedbacks!**
>
> 1. The use of GBDTs for moderator discovery indeed captures nonlinear interactions; but also results in discontinuous, piecewise-constant outputs. This discretization may not be able to capture smooth moderation patterns, limiting method generalization.
>
> We thank the reviewer for raising this point.
>
> While it is correct that individual tree leaves produce piecewise-constant outputs, this property does not limit the ability of GBDTs to capture moderation structures because our moderation analysis is performed on SHAP interaction values, not the raw tree output. Moreover, the moderator coefficients $\beta_3$ are obtained by an additional regression-based assessment step.
>
> 2. TreeSHAP assumes feature independence; this may create spurious moderators or omit true ones.
>
> We agree that the independence assumption of TreeSHAP is an important practical consideration. However, TreeSHAP’s independence assumption applies only to the *marginalization distribution*, not to the learned model. The GBDT fully captures the true joint dependence among features during training. TreeSHAP is merely used to decompose the model output, not to infer causal structure from raw data. The moderating effect is modeled by the regression-based assessment.
>
> 3. ShapMod-1st outperforms ShapMod (based in Eq. 6) in three of four datasets, raising a question about the need for SHAP interaction terms in Eq (6)?
>
> We appreciate this insightful observation and clarify the roles of the two variants:
>
> The ShapMod-1st uses marginalized first order SHAP values, therefore the interactions are absorbed into each main effect.
>
> The ShapMod used second order SHAP interactive values, which explicitly captures the contribution of interaction terms.
>
> In most cases, their performance difference should be negligible, as demonstrated in the experiments. We explicitly formulate interaction terms in ShapMod-1st because we aim to explicitly demonstrate that the interaction terms are the key factor to moderator discovery.
>
> 4. Experiments are small ; generalizability unclear.
>
> The experiment sizes are typical for social science datasets. Moderation discovery is inherently an interpretability-focused problem. In social science datasets sample sizes of 1k–10k are normal.

---

### Official Review · Reviewer_jHLQ · 2025-11-02

**Soundness:** 2
**Presentation:** 3
**Contribution:** 3
**Rating:** 4
**Confidence:** 2

**Summary:**

This paper focuses on the problem of automatically identifying moderator variables. The authors propose a three-step approach that utilizes SHAP interaction values to train a predictive model, evaluate moderation effects, and identify significant moderators. The effectiveness of the proposed framework is validated on both synthetic datasets and real-world dataset.

**Strengths:**

1. The paper is well-written and clearly organized.

2. The authors provide a good review of related work.

3. The experimental results on the Medicare vaccination dataset reveal an interesting and potentially novel pattern, which could bring useful insights to this community.

**Weaknesses:**

1.  The baseline methods (e.g., Lasso and ANOVA) seem a bit outdated. I am not an expert in this particular subarea, but if there are newer or more advanced baselines available, including them could make the evaluation more convincing.

2. The proposed framework currently lacks a theoretical guarantee. It would be helpful to discuss under what conditions the identified moderator can be regarded as the most important one. Adding some theoretical insights or sensitivity analysis could further strengthen the contribution.

**Questions:**

See Weaknesses.

---

> ### Author Response · Authors · 2025-12-04
> **Thanks for the valuable feedbacks!**
>
> We thank the reviewer for the thoughtful feedback. Below we provide clarifications on the raised concerns.
>
> 1. The baseline methods (e.g., Lasso and ANOVA) seem a bit outdated.
>
> We appreciate the reviewer raising this point. We would like to clarify that Automatic Moderator Discovery is a distinct and relatively unexplored task within the Machine Learning community, differing significantly from standard Interaction Detection.
> As discussed in our Related Work (Section 2.2), the most relevant are general interaction detection methods (such as Integrated Hessians (Janizek et al., 2021), Archipelago (Tsang et al., 2020), and FISC (Li et al., 2023)) focus on identifying that an interaction exists . However, these methods typically stop at the "general interaction level" and do not provide a fine-grained analysis of the conditional relationship.
>
> The core of moderation analysis is determining *how* the effect of a predictor ($X$) on the outcome ($Y$) varies systematically with the level of a moderator ($Z$). Existing modern interaction detection methods do not explicitly model this directional conditioning (i.e., quantifying $\beta_3$ ). Consequently, classical methods like Hierarchical Lasso and ANOVA, which explicitly construct and test product terms ($X \times Z$), remain the closest functional baselines for this specific task.
>
> To our knowledge, there is no existing "state-of-the-art" ML framework specifically designed for this type of automated *moderator* discovery, which is precisely the gap our work aims to fill.
>
> 2. Theoretical Guarantees and Sensitivity Analysis
>
> We agree with the reviewer that theoretical guarantees are desirable. However, we respectfully argue that defining a "true" moderator is not solely a data distribution problem, but one that is intrinsically tied to domain theory and causal logic, which mathematical formulations alone cannot fully resolve.
>
> Mathematically, an interaction term is symmetric ($Y \propto \beta X Z$). The data alone cannot distinguish whether $Z$ moderates $X$, or $X$ moderates $Z$. For example, consider $X$=*Study Time*, $Z$=*Sleep Quality*, and $Y$=*Exam Score*. A model might find a strong interaction, but only domain logic determines that *Sleep Quality* moderates the efficiency of *Study Time*, not vice versa. In many cases, an interaction may exist without a meaningful moderating relationship.
>
> Because of this semantic ambiguity, our framework is designed as a **Discovery Tool**  rather than a causal oracle. We provide statistical guarantees regarding the significance of the interaction candidates (via statistical significance), but the final theoretical validation of the "moderator" role must involve the domain user.
>
> We hope these clarifications address your concerns. We believe our framework serves as a vital bridge, enabling ML to move beyond prediction toward verifying and generating scientific theory.

---

### Meta-Review · Area_Chair_uv4r · 2026-01-07

**Summary:**

The paper proposes a three-step framework for the automatic discovery of moderator effects (conditional relationships between variables) using SHAP interaction values derived from Gradient Boosted Decision Trees (GBDTs). The framework involves training a GBDT, computing pairwise SHAP interaction values, and then regressing these values back onto the features to identify and rank statistically significant moderators.

**Reviewer Concerns:**

Reviewers consistently noted that the method is a heuristic pipeline of existing tools (TreeSHAP and OLS) rather than a novel algorithm. Significant technical concerns were raised regarding the lack of theoretical guarantees, the discretization artifacts of GBDTs, and the conceptual leap from symmetric interaction values to directional moderation effects. Furthermore, the evaluation was deemed insufficient due to a lack of modern baselines and sensitivity analyses.

**Reviewer Scores:**

The scores were predominantly on the negative side: 2, 2, 2, 4, and 4. While some reviewers appreciated the clarity and the potential utility in social sciences, the consensus leaned toward rejection.

---

### Decision · Program_Chairs · 2026-01-26

Reject